# Effect of Dietary Hemp Cake Inclusion on the In Vivo and Post Mortem Performances of Holstein Veal Calves

**DOI:** 10.3390/ani12212922

**Published:** 2022-10-25

**Authors:** Sheyla Arango, Nadia Guzzo, Emiliano Raffrenato, Lucia Bailoni

**Affiliations:** Department of Comparative Biomedicine and Food Science (BCA), University of Padova, Viale dell’Universitá 16, 35020 Legnaro, PD, Italy

**Keywords:** hemp cake, veal calves, meat quality, fatty acid profile

## Abstract

**Simple Summary:**

Hemp is a multifunctional crop with seeds containing high levels of polyunsaturated fatty acids (PUFA), such as alpha-linolenic acid (ALA), that may transfer to enrich animal-based products. This experiment was carried out on 52 male Holstein calves that were divided into two homogeneous groups and fed two concentrates containing 0 and 3% of hemp cake. Results showed that hemp cake inclusion increased the growth in the first period of the trial and improved some carcass parameters. Cooking weight losses and shear force of meat were negatively affected by the addition of hemp cake in the concentrate. The expected transfer of ALA in meat was not detected. The use of hemp cake in the diet of veal calves could be considered as a safe ingredient, but the optimal dosages will need to be further investigated.

**Abstract:**

Fifty-two male Holstein veal calves were divided into two homogeneous groups receiving two isoenergetic and isonitrogenous concentrates without (CTR group) or with 3% of hemp cake (HC group). The trial lasted for 171 days. All the calves were weighed five times during the trial. At slaughtering, carcasses were weighed and measured. Meat quality was determined on the *Longissimus dorsi* muscle. Average daily gain in the first period of the experiment (0–80 d) and dressing percentage and rump width of the carcasses were higher in HC group. Cooking weight losses and shear force were higher in the meat of the HC group while color parameters were similar in the two experimental groups. Unexpectedly, the alpha-linolenic acid content of meat was lower in the HC group. In conclusion, hemp cake can be considered an interesting ingredient in the concentrate used for the production of veal calves, but further studies will be needed to determine a suitable dosage in order to improve the nutritional quality of meat (i.e., the n-3 fatty acids content) without negative effects on physical characteristics.

## 1. Introduction

Hemp (*Cannabis sativa* L.) is a multifunctional crop [1]. Around 60% of the total worldwide hemp production takes place in Europe [2]. Hemp is grown for various applications, mostly to obtain fiber for making light-weight papers, insulation material and biocomposites [3]. Hemp seeds are used predominantly in human [4] and animal nutrition [5,6,7,8,9] and they can be cold-pressed to obtain oil and hemp cake. Both co-products are used in animal feed [3,8]. Hemp cake is an optimal source of protein (on average 34.3 ± 2.1% on DM basis) and energy, considering the high percentage of residual oil (12.7 ± 2.8% on DM basis) [6,10]. In addition, the oil shows a good fatty acid profile, with a high content of PUFA (on average 75% of total fatty acids), especially linoleic acid (LA; 18:2 n-6) and linolenic acid (ALA, 18:3 n-3) [6]. 

Essential fatty acids (FA) play an important role in human health and a valuable source of these nutrients is meat. Unfortunately, meat consumption seems to be associated with two of the major chronic diseases in the Western world: cardiovascular disease and colon cancer, probably because of the saturated fatty acid (SFA) content [11]. In fact, the fatty acid profile of meat is primarily monounsaturated fatty acids (MUFA) (45–50%) and SFA (45–55%), and very low amounts of PUFA (10%) [12,13]. In order to reduce the risk of cardiovascular diseases, guidelines for decreasing total FA intake, and replacing SFA with PUFA, especially those of the n-3 series, has been recommended by the World Health Organization [14]. This is why meat consumers not only consider a low level of fat content mandatory, but also see the fatty acid composition of any meat as an important point of consideration [15]. 

To guarantee the production of healthier meat, various investigations have been made over the last decade. One method that has increased in popularity is manipulating the fatty acid profile, specifically increasing the n-3 fatty acid content of beef to increase its appeal in a healthy diet [12]. Knowing that the three major factors that influence the FA composition of beef are the age, breed and diet of the animal [11], many researchers have chosen to work on the animal’s diet because fatty acid metabolism in the rumen depends primarily on this [16]. Despite extensive ruminal modification, diets high in specific unsaturated fatty acids can increase the concentration of FA in deposited fats [17]. These diets include the dietary supplementation of oilseeds [12] that can reduce the ruminal modification of FAs and may therefore increase the resemblance between dietary and tissue FAs [17]. Supplementation of the diet with PUFA-rich oilseeds such as hemp has led to increases in conjugated linoleic acid isomer (CLA) levels [16] and some beneficial effect of hemp supplementation on the fatty acid composition has been shown in chicken [18] and duck meat [19]. Moreover, one study reported that hemp seeds can favorably alter carcass fat by increasing levels of CLA and n-3 fatty acids in beef without negatively affecting the animal performance [17]. No other studies have been carried out on meat with hemp to increase knowledge regarding this source of PUFAs. 

The use of hemp in healthy meat production may also have an advantage, as it is rich in an antioxidant called tocopherol [20]. Producing beef with meaningfully enhanced concentrations of biohydrogenation products can alter meat quality because PUFAs are more susceptible to oxidation and can cause surface discoloration [12]. This is an issue, as meat color is most often used as an indicator of product freshness and quality by consumers [12]. Therefore, the presence of tocopherol may also help to keep meat color parameters unaltered.

Since altering the nutritive value of meat has become the focus of a number of producers [12], new sources of oilseeds need to be tested for this purpose. The objective of this study was to evaluate dietary hemp cake inclusion in Holstein veal calves on in vivo and post mortem performance, with particular attention on meat quality and fatty acid profile.

## 2. Materials and Methods

### 2.1. Ethics Statement

All experimental procedures were carried out according to Italian law on animal care (Legislative Decree No. 26 of 14 March 2014) and approved by the ethical committee at the University of Padova (approval number 38/2022).

### 2.2. Animals and Diet

The experiment was conducted on a farm located in the province of Treviso, in the north-east of Italy. A total of 52 Holstein male calves with 25 ± 13 d of age were chosen. The calves were randomly assigned, based on their initial body weight (BW), into two homogeneous groups called control (CTR) and hemp (HC). They were housed in one room divided into 10 pens (5 per group). Each pen housed 4 to 6 calves, with a space allowance of 1.8- m²/calf. Moving from pen 1 to pen 5, the average weight or size of the calves increased but remained homogeneous inside the pen in order to avoid hierarchical or aggressive behavior. The pens had wooden slatted floors. There were 3 feeders per group, every two pens sharing one and the last pen having an individual feeder. Environmental temperature in the house was controlled and maintained by an extractor fan system at 22 °C. The experimental period lasted for 171 days.

The calves from the two experimental groups were fed the same commercial milk replacers during the whole trial. Two milk replacers that met the calves’ nutrient requirements during the fattening period were used. The first milk replacer (Denkaveal Start, Denkavit Italiana S.r.l, Brescia, Italy) was used during the adaptation period (24 days) and then in the first 7 days of the experimental period. During the next 29 days, the first milk replacer was mixed in a 50:50 ratio with the second milk replacer (Sharmel Unico Light S, Frabes S.p.A., Brescia, Italy). In the following part of the experiment (135 days), calves received only the second replacer. The daily dose of milk was delivered in two equal meals at 06:30 and 18:30. The daily amount of milk powder and its concentration in the liquid diet increased throughout the trial from 830 to 2120 g per calf per day for the two experimental groups. 

Two isoenergetic and isonitrogenous concentrates with conventional protein sources and with 3% of hemp cake were formulated, respectively, for the control group (CTR: 26 animals; 5 pens) and hemp cake group (HC: 26 animals; 5 pens). The hemp (*Cannabis sativa* L.) used in the trial was Futura 75 variety, cultivated in Ferrara (Italy). The oil was extracted from seeds using a cold process by the company Vergavara Lab (Rossano Veneto, Vicenza, Italy). The residual hemp cake was used by the feed company Italfiocchi Monfort S.r.l. (Castelfranco Veneto, Treviso, Italy) to produce the HC concentrates. The chemical composition of hemp cake is reported in Table 1.

During the adaptation period (24 days) all the animals received the control diet through a concentrate 1 (Denkaveal Avance PI Integrato Fiocco +P DP, Fanin S.p.A., Vicenza, Italy). The experimental period was divided into two phases. In the first phase (61 days), the CTR group was fed with concentrate 1 and the HC group received concentrate 2 (Avance Mix Fiocco Fiber 5% Canapa, Italfiocchi Monfort S.r.l., Treviso, Italy). In the second phase (110 days), the CTR group was fed with concentrate 3 (Avance Mix Fiocco Omega Fiber 5%, Italfiocchi Monfort S.r.l., Treviso, Italy) and the HC group with concentrate 4 (Avance Mix Fiocco Omega Fiber 5% Canapa, Italfiocchi Monfort S.r.l., Treviso, Italy). The inclusion of the hemp meal was made by substituting 6 and 4.5% of lupin seeds in the first and second phase, respectively. The concentrates were distributed ad libitum. During the trial, the amount of concentrate increased on average from 943 to 3922 g/d. The DM intakes of each group were recorded daily. For each pen, the total DM intake (milk replacer and concentrate) was calculated. 

The chemical composition and fatty acid profile of milk replacers and concentrates are reported in Table 2 and Table 3 respectively.

Drinking water was available ad libitum.

### 2.3. Animal Monitoring

Upon arrival, calves were vaccinated, treated for external and internal parasites and checked by a veterinarian to verify their health status. Also, all the animals were supplemented orally with 1 g Fe/day (Ferro tonic, Daily Manufacturing, Rockwell, NC, USA) from day 11 to day 20 of the trial.

During the experiment all calves were individually weighed five times (at day 1, 38, 80, 122 and 171) in order to assess the average daily gain (ADG). Dry matter efficiency was estimated as daily gain/DM intake ratio. 

The health status of the calves was monitored through blood analysis at the beginning, middle and end of the experimental period. Blood samples were taken from all the calves by jugular vein puncture before the morning meal. Heparinized vacutainer tubes (FL MEDICAL SRL, Padova, Italy) were used for assessing plasma hemoglobin and iron concentration according to procedures PDP ACC 075 2022 Rev.1 and PDP ACC 043 2018 Rev.1, respectively, of Istituto Zooprofilattico Sperimentale delle Venezie (Legnaro, Padova, Italy). 

At the end of the trial, 51 calves were slaughtered in an authorized, commercial slaughterhouse (Bencarni S.p.a., Nogarole Rocca, Verona, Italy) following the recommendations of the European Council regarding the protection of animals at the time of killing. Around 30 min after slaughter, the hot carcass weight was measured and the individual dressing percentage was calculated. The thigh length, rump width and pH were measured on each hot carcass. Then, the carcasses were aged in a controlled room for 7 days at a temperature of 0 °C–4°C.

### 2.4. Chemical and Technological Analyses

Samples of hemp cake, milk replacers and concentrates were collected at the beginning and at the end of the experiment and analyzed for dry matter, protein, lipids and ash according to the association of official analytical collaboration [21]. Neutral detergent fiber (NDF), inclusive of residual ash, was determined with α-amylase using the Ankom220 Fiber Analyzer (Ankom Technology, Macedon, NY, USA). Acid detergent fiber (ADF), inclusive of residual ash, was determined sequentially after NDF determination [22]. Starch content was determined after hydrolysis to glucose by liquid chromatography [23].

To quantify the iron content, a mineralization was performed using a Start D microwave digestion system (Milestone Srl., Sorisole, BG, Italy). The samples were digested with 7 mL of HNO_3_ super pure and 2 mL of H_2_O_2_ 30% by bringing them up to 200 °C in 15 min and remaining at that temperature for a further 15 min. The digested samples were diluted with deionized water into 25 mL volumetric flasks. The concentration of iron was measured by ICP-MASS Spectrometry (EPA 6020A 2007).

To evaluate meat quality characteristics, 7 days after slaughter, 24 samples of *Longissimus dorsi* muscle (12 for each experimental group) were taken from the fifth–sixth rib. The sample was vacuum-packaged and stored at 4 °C in a chilling room for 6 days. After this aging period, the meat samples were frozen and kept at −20 °C until analysis.

Meat chemical analyses considered moisture, intramuscular fat (IMF), protein content and iron concentration (see above). Technological analyses considered pH, color, cooking losses and shear force. The pH was measured by a portable pH-meter provided with a 5050T electrode (Hach Lange S.r.l., Milan, Italy). Meat color was measured with a CM-600d spectrophotometer (Konica Minolta Inc., Tokyo, Japan) on samples following the AMSA method [24]. Samples of meat were measured by scanning 3 different spots and color data were expressed according to the CIE L*, a* and b* system. The instrument was calibrated with a white standard plate before measurements. Weight cooking losses were determined on 2 cm thick steaks heated in a water bath at 75 °C for 60 min and cooled in running tap water for 15 min. After cooking and before opening the bags, each sample was tempered at room temperature to drain liquid. 

The cooking losses were calculated using the formula:(weight of raw meat − weight of cooked meat)/weight of raw meat × 100.

The instrumental measurement of meat tenderness was carried out using a Lloyd Instrument LS5 shear force meter (AMETEK Inc., Thurmaston, LE, UK) on five cylindrical core samples of cooked meat of 1.4 cm in diameter. The measurement was recorded and calculated with the instrument software Newygen Plus 3 as the peak yield force in N, required to shear, at a 250 mm/min crosshead speed, perpendicularly to the direction of the fibers on five cylindrical cross-section (4 × 8 × 1 cm dimension) replicates from each sample [25].

### 2.5. Fatty Acid Profile of Feed and Meat

The fatty acid profile of samples (feed and meat) was determined by a preliminary extraction of fat using an accelerated solvent extraction (ASE 200, Dionex Corp., Sunnyvale, CA, USA) with petroleum ether. The GC with flame-ionization detector (Agilent Technologies Inc., Shanghai, China) had a temperature of 300 °C and was equipped with two columns in series and with a modulator (Agilent G3486 A CFT), an automatic sampler (Agilent 7693) and specific machine software (Agilent Chem Station) were used to determine the concentration of the single fatty acids. This instrument was chosen because the double column allows for separating and identifying each FA on a 2-dimensional basis [26] (Pellattiero et al., 2015). The first column was a 75 m × 180 μm (internal diameter) × 0.14 μm film thickness column (23,348U, Supelco, Bellefonte, PA, USA), the second was a 3.8 m × 250 μm (internal diameter) × 0.25 μm film thickness column (J&W 19091-S431, Agilent Technologies). The first and the second column used H2 as carrier gas at a flow rate of 0.25 and 20 mL/min, respectively.

The concentration of each fatty acid was expressed as g/100 g, considering 100 g as the total of areas of all FAME identified.

### 2.6. Statistical Analysis

The normal distribution of all the variables included in the dataset were tested and then submitted to an ANOVA within PROC GLM (SAS Inst. Inc., Cary, NC, USA). 

For performance parameters, data were analyzed using a nested design in which each calf was an experimental unit, and the effect of the pen was considered.
Y_ij_ = µ + D_i_ + D_i_ (β_j_) + ε_ij_
where Y_ij_ are the observations, µ the overall mean, D_i_ the effect of the diet (i = 2), β_j_ the effect of the pen and ε_ij_ is the random residual.

For blood parameters, a mixed model using the period (3 periods) as an effect was used. Data of meat parameters were submitted to a completely randomized design. 

For all the variables, the comparisons between LS means were performed using the Tukey test and differences were considered significant at *p* < 0.05.

## 3. Results

### 3.1. Animal Performance 

One calf of the CTR group died during the trial. The cause of death was due to a respiratory problem. For this reason, this animal was discarded from the data set. 

The use of hemp cake in veal calves’ diet did not affect (*p* > 0.05) the final body weight or the average daily gain of the whole trial. Similarly, DM intake (milk and concentrate) and feed efficiency for both groups were not affected (Table 4).

The “pen” effect was significant for all the variables reported in Table 1, but this is an expected result considering that the distribution of calves in the five pens, within each experimental group, was based on the initial BW, as reported above.

Considering the pattern of growth, the maximum value of ADG was observed from day 81 to 122 of the trial for both experimental groups (1830 and 1719 g/d for CTR and HC groups, respectively; *p* > 0.05). Only during the first phase were the differences of ADG between CTR and HC group statistically significant, showing higher values for the HC group (1017 vs. 1141 g/d, respectively, from day 1 to 38 and 1417 vs. 1533 g/d from day 39 to 80; *p* < 0.05) (Figure 1).

### 3.2. Animal Health Status 

The incidence of respiratory problems was very low in both groups (19.2 vs. 11.1% in CTR e HC group, resp.) and few cases of gastrointestinal problems were observed in the HC group (7.41%).

In the blood samples taken at 5, 62 and 128 days of the experiment, the iron concentration (Figure 2a) was similar in the two groups (on average 84.95 μg/dL). The hemoglobin concentration (Figure 2b) was lower and more constant during the fattening period in the HC group in comparison to CTR group (9.64 vs. 9.33 g/dL; *p* < 0.05).

### 3.3. Meat Characteristics

Even though the carcass weight was similar for both groups (162.8 vs. 164.6 kg for CTR and HC group, resp.; *p* > 0.10), the dressing percentage was higher (49.9 vs. 51.5%; *p* < 0.001) in the HC group. In addition, the conformation of carcasses was better in the HC group, considering, in particular, the rump width (37.20 vs. 38.38 cm; *p* > 0.05). The pH of carcasses was not affected (*p* > 0.05) by the hemp cake inclusion.

The color parameters (L, a, b, H, C,) of meat obtained by the Minolta spectrophotometer were very similar between experimental groups (Table 5). The lightness of meat in the HC group tended (*p* = 0.10) to be higher than that of the CTR group. The weight cooking losses and shear force showed significantly higher values in the HC group (29.49 vs. 31.13%; *p* < 0.05 and 25.79 vs. 36.18 N; *p* < 0.001 respectively). 

Regarding the chemical composition of the *Longissimus dorsi* muscle, no differences (*p* > 0.05) were observed for water, intramuscular fat (IMF), protein and iron content.

### 3.4. Fatty Acid Profile of Meat

For both experimental groups, the most represented SFA in meat was palmitic acid (C16:0, on average 26.5% of the total FA) followed by stearic acid (C18:0, on average 12.3% on the total FA) (Table 6). The most abundant unsaturated fatty acid proved to be oleic acid (C18:1 n9), which was higher in the HC group as compared to the CTR group (38.9 vs. 40.3% of total fatty acids; *p* < 0.05). The inclusion of hemp cake in concentrate caused a decrease in the n-3 fatty acids, mainly represented by alpha linolenic acid (ALA, C18:3 n3, 0.45 vs. 0.36% of the total FA; *p* < 0.05). The n-6:n-3 ratio, considered an important nutritional index, was higher in the HC group (16.8 vs. 19.9 for CTR and HC group, respectively; *p* < 0.05).

## 4. Discussion

To our knowledge, no scientific papers have been published on the use of hemp cake in the diet of veal calves.

### 4.1. In Vivo Performance

The results of this experiment showed a good growth and fattening during the whole (171 d) trial for Holstein calves fed concentrate with the inclusion of hemp cake. The replacement of traditional protein and energy sources with the product obtained as residual material of the cold hemp oil extraction allows high in vivo performance to be obtained. The calves receiving hemp cake in the concentrate reached the slaughter weight at the same time as the animals fed the control diet or conventional diets [27].

In the first phase of the fattening period (1–80 day), the daily gain of calves receiving hemp cake was significantly higher than that of the control group. It could therefore be hypothesized that hemp cake is more efficient in sustaining growth in the first phase of life when protein daily gain begins and prevails over fat gain. This hypothesis is supported by the recent results of hemp seed protein in human nutrition reported by Farinon et al. [4]. The authors recognized that whole hemp seed can be considered not only a rich-protein source (25.6% of crude protein on DM), higher than or similar to other protein-rich products (i.e., flaxseed, 20.9 and lupin seed, 30.5% of crude protein on DM) [28] but also a good source of essential amino acids. The amino acids profile of hemp seed (Finola variety) is similar to those of casein in milk except for lysine, which is the first limiting amino acid of hemp seed [4,29]. In this experiment, the 6% of hemp cake was included in the concentrate in replacement of white lupin seed in the first phase. As reported by Mattila et al., the lysine content of hemp seed is lower than that of white lupin seed (3.30 vs. 5.80 g/100 g of protein), however, on the contrary, methionine level is very high in hemp cake (2.19 vs. 0.80 g/100 g protein) [28]. In the first period, veal calves ingest a high quantity of milk replacer, based on whey derivatives, which could have satisfied the requirements of lysine for growth. Therefore, it is possible to hypothesize that the hemp cake inclusion could be useful, in this phase, for providing a suitable quantity of the secondary limiting amino acid, that is, methionine. 

In the second phase of the fattening period, the values of daily gains of calves were similar in the two experimental groups, probably owing to a compensatory growth of subjects receiving the concentrate without hemp cake. In addition, the percentage of hemp cake inclusion in the concentrate of the HC group during the second period was lower (4.5% instead of 6%). Also, the amino acid requirements change in relation to the age and weight of calves following the modifications of the chemical composition of daily gain.

Considering the whole experiment, the DM intake of concentrate is similar for both groups. This indicates that the palatability of the concentrate containing both 6% and 4.5% of hemp cake was high for veal calves. Hessle et al. found an increase in total DM intake when a high quantity of hemp cake (1 kg/d) is included in place of soybean meal in the diet of dairy calves (from 96 to 250 kg BW) fed mixed rations based on grass/clover silage and rolled barley [30].

No differences in feed conversion ratio between the two experimental groups were observed in this experiment. Feed conversion ratio was similar to that reported recently by Van Gastelen et al. in an experiment with Holstein-Friesian calves fed different solid feed mixtures, but considering a short experimental period (from 13 to 17 weeks of age) [31]. 

Regardless of the feeding treatment, the health status of the calves was considered optimal as shown by the normal levels of hemoglobin and iron. Special consideration for these two parameters is given in this type of breeding because of the regulations that state a minimum hemoglobin level of 7.25 g/dl [32]. The hemp supplementation significantly decreased the calves’ hemoglobin in blood, but they were never below the recommended level. The iron content in blood was not affected by the treatment, which means that this level of hemp supplementation can be considered safe for the animal’s welfare. In conclusion, the iron content provided by the hemp cake in the concentrate allows a good state of health and welfare of the calves without compromising the color of the meat, which must be pale as shown below.

### 4.2. Post Mortem Performance

Hemp cake inclusion did not affect the carcass weight, but the dressing percentage was higher for the HC group. Dressing percentage values (49.91 and 51.47% from the CTR and HM group, resp.) in this trial are lower than the average of 54.8% calculated from previous studies in Holstein calves slaughtered at weights between 210 to 280 kg [27,33,34,35]. Although the supplementation (full-fat hemp seed vs. hemp cake) and the category of animals (steers vs. calves) are different, these results are consistent with a previous study of Gibbs et al. in which the effects of full-fat hemp seed were not significant in steers fed with a barley-based finishing diet [17]. However, rump width was significantly improved with hemp inclusion. This parameter measures the carcass in terms of size and describes the animal growth better than conventional methods of weighing [33]. 

Regarding the instrumental color variables, the results obtained in this experiment are in agreement with those reported by Brugiapaglia et al. in white veal calves (L* 53.37, a* 6.91, b* 15.15, C* 16.76, H* 65.75) [36], showing a very light pale color, as it should be, due to high L* values together with low a* and high b* values. For parameters such as redness (a*) and color vividness (C*), the values 12 and 16, respectively, are considered as thresholds for visual discoloration in beef. In this case, as we are dealing with white veal meat, it was normal to find values below these two suggested for red beef. Also, as the veal industry depends strongly on lean color, and whiter graded carcasses command greater value [34], and knowing that hemp is a source of PUFA, its addition in the diet could have made the meat more susceptible to oxidation, thus causing surface discoloration [12]. However, according to our results, all these values were not significantly affected by the diet and were similar to those observed by Brugiapaglia et al. for white veal meat of Holstein male calves [36].

Cooking loss and shear force were affected by the experimental diet. Hemp cake supplementation significantly increased cooking loss of the *Longissimus dorsi* muscle. The values found in this study were near the 28.6 [27] and 30.9% [14] reported in the literature for Holstein calves. The tenderness of the meat is the most important palatability characteristic for consumers [12,36]. Hemp cake supplementation negatively affected this parameter by increasing the shear force of the meat. Even though the values found in this study were higher than the 23.94N reported for white veal calves [27], this negative effect does not agree with the fact that intramuscular fat positively influences meat tenderness and, as both groups had similar intramuscular fat values, they should have had similar values.

Water, intramuscular fat, protein and iron content in the *Longissimus dorsi* muscle were not affected by the hemp cake supplementation. Moisture and protein values were similar to the ones found in the literature [14,37]. Intramuscular fat plays an important role in palatability and levels between 3.0 and 7.3 g/100 g of muscle have been generally considered acceptable for consumers in terms of visual quality and health concerns [14]. Considering this, the intramuscular fat (3.71 g/100 g muscle) values in this study were satisfactory but were higher than the 1.94% reported in Holstein bulls fed with concentrate [38] and the 0.65% reported in the same muscle of Holstein calves [34]. Owing to the young age of the animals, the diet and the fact that we are dealing with white veal meat, the iron concentration in the meat was very low (2.83 for the CTR and 2.95 mg/kg muscle for the HC group) and no significant difference between the two groups was observed. Values are near the ones reported in the literature for Holstein male calves slaughtered at 6 months with 3.77 mg of iron/kg of meat [36].

For both groups, and supported by the literature [14,39], oleic acid (C18:1) was the most abundant FA, followed by palmitic (C16:0) and stearic acid (C18:0). Our data showed that the HC group had higher proportions of oleic acid than the CTR group even though the HC diet had a lower quantity of this FA. On the one hand, similar values were reported for the oleic acid content, such as 38.64 in Holstein rib samples at a slaughter weight of 401–500 kg [15] and 39.64 in beef from steers fed with 9% of full-fat hemp seed [17], while on the other hand, literature findings state that feeding hemp oil and seed reduces levels of oleic acid in chicken breast meat [18] and also in beef [17]. The other two major FAs, palmitic and stearic, did not show any difference between treatments.

For PUFA, linoleic acid (C18:2) tended to predominate, but it was in the linolenic acid (C18:3) that a difference was found. It is well known that n-3 FAs such as linolenic acid have been recognized as being beneficial for human health. Values of 0.34 [39] and 0.53 g/100 g of total FA [15] were reported on male Holstein rib samples. In fact, including flaxseed [12] and hemp seed in diets has been shown to increase linolenic acid content within beef cuts [17]. However, in this trial the significance of a higher level in the CTR than the HC group was unexpected. This could be due to the low content of ALA (12.82 g/100 g of total FA) in the hemp cake used, as the average range reported is between 14.62 and 19.10 g/100 g of total FA [6]. For instance, the only experiment using hemp to improve the fatty profile of meat used full-fat hemp seed with an ALA content of 24.6 and found an improvement of +51.45% of this FA in the meat [17]. Beef is a natural source of CLA, and it is derived from dietary PUFA [16]. Supplementation of the diet with PUFA-rich oils or seeds has led to increases in CLA, but as normal concentrations are low the increases are not appreciable [16]. Typical concentrations are less than 1% of total FA [16], as was found in this study.

As regards the FA classes, there were no significant differences between the treatments. The FA distribution of white veal is similar to other meat products and was composed primarily by MUFAs (47.2%) and SFAs (45.3%) with very low amounts of PUFA (7.5%). The sum of n-3 FA, which was basically the quantity of ALA, was lower in the HC group (0.36) and it is also lower than the 0.96 reported for intramuscular fat in the longissimus muscle of German Holstein bulls fed with concentrate [39]. The lower n-6:n-3 ratio of the meat in the CTR group (16.77) in the current study indicates a healthier nutritional profile than the HC group (19.85). Both results were higher than the recommendation of 4 by the UK Department of Health and also seemed high in comparison to that of 2.1 given as an average for cattle meat [40].

## 5. Conclusions

Using hemp cake as an ingredient in the concentrate of Holstein veal calves can be considered safe since it did not affect the calves’ health status or the in vivo performance parameters. However, average daily gain in the first period of the experiment and dressing percentage had an effect on the muscular growth with the hemp cake inclusion. Moreover, this inclusion did not change the meat color, which remained pale (low values of redness and yellowness) and for this reason still well accepted by consumers. Other approaches should be considered in order not to alter meat quality parameters such as cooking loss, shear force and the n-6:n-3 ratio. The enrichment of n-3 fatty acids of meat, obtained by the calves receiving hemp cake, was not successful due to a possible low percentage of hemp inclusion in the diet. Further experiments using higher doses are suggested to improve the fatty acid profile of meat.

## Figures and Tables

**Figure 1 animals-12-02922-f001:**
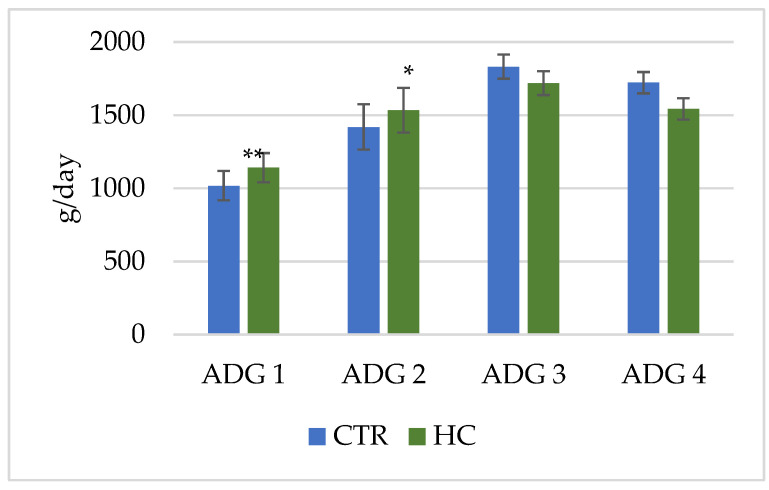
Least square means of average daily gain for control (CTR; blue bars) and hemp cake (HC; green bars) groups of veal calves during the different phases of the experiment (ADG1; 1–38 days, ADG2; 39–80 days, ADG3; 81–122 days, ADG4; 123–171 days). Differences between treatments within phase have been reported when significant: (*) *p* < 0.05, (**) *p* < 0.01.

**Figure 2 animals-12-02922-f002:**
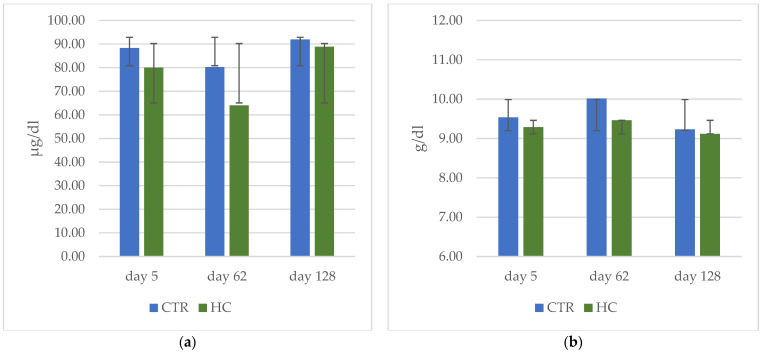
Health status parameters for CTR (control group) and HC (Hemp group) taken in three different periods: Initial, at the beginning of the trial; Intermediate, in the middle of the trial and Final, at the end of the trial. (**a**) Iron values (**b**) Hemoglobin values.

**Table 1 animals-12-02922-t001:** Chemical composition (% on DM), iron content (mg/kg) and fatty acid profile (g/100 g of total FA) of the hemp cake.

Hemp Cake	
Chemical composition	
DM	92.40
Crude Protein	28.17
Lipids	8.70
Ash	6.19
NDF	50.91
ADF	37.38
ADL	11.54
AIA	0.72
Iron	168.80
Fatty acids	
C14:0	0.07
C16:0	8.28
C18:0	3.04
C18:1n-9	15.79
C18:2n-6	56.20
C18:3n-3	12.82
Total SFA ^1^	13.33
Total MUFA ^2^	17.46
Total PUFA ^3^	69.21
n-6/n-3	4.4

^1^ SFA, saturated fatty acids; ^2^ MUFA, monounsaturated fatty acids; ^3^ PUFA, polyunsaturated fatty acids.

**Table 2 animals-12-02922-t002:** Chemical composition (% on DM) and iron content (mg/kg) of milk replacers and concentrates.

			CTR		HC	
Item	Milk Replacer 1 ^1^	Milk Replacer 2 ^2^	Concentrate 1 ^3^	Concentrate 3 ^4^	Concentrate 2 ^5^	Concentrate 4 ^6^
Dry matter	94.05	93.98	89.95	90.69	90.04	90.48
Crude protein	24.27	18.71	13.12	13.02	13.87	13.51
Lipids	18.31	15.77	3.34	2.36	2.25	2.37
Ash	8.02	6.72	4.32	4.11	4.65	4.28
Starch	nd	nd	38.62	40.70	41.95	39.62
NDF	nd	nd	18.60	24.09	22.92	27.22
ADF	nd	nd	7.99	10.61	10.45	13.20
Iron	19.26	16.77	63.34	64.25	40.72	68.99

^1^ Ingredients of Milk replacer 1: proteins of whey milk powder, whey powder, delactosed whey powder, animal fats (pork and bovine), wheat protein, vegetal oils (coconut, soy), wheat meal, protein of extruded peas, protein concentrate of soybean seed, calcium carbonate, monopotassium phosphate, magnesium oxide, fatty acids of tall oil, 1,2–propanediol. Vitamin and mineral content (per kg): 12,450 IU of vitamin A; 3900 IU of vitamin D3; 80 IU of vitamin E; 8 mg of Fe; 0.5 mg of I; 10 mg of Cu; 30 mg of Mn; 40 mg of Zn; 0.1 mg of Se. ^2^ Ingredients of Milk replacer 2: whey powder, animal fats (pork and bovine), delactosed whey powder, wheat protein, wheat meal, vegetal oils (coconut, soy), protein concentrate of soybean seed, protein of extruded peas, dextrose, calcium carbonate, monopotassium phosphate, magnesium oxide, 1,2–propanediol. Vitamin and mineral content (per kg): 12,500 IU of vitamin A; 2000 UI of vitamin D3; 80 IU of vitamin E; 0.5 mg of I; 3 mg of Cu; 10 mg of Mn; 80 mg of Zn; 0.1 mg of Se. ^3,4^ Ingredients of Concentrates 1 and 3: corn flakes, corn grain, wheat straw, barley flakes, white lupin flakes, barley seed, white lupin seed, pea flakes, corn gluten meal, calcium carbonate, sodium chloride and vitamins. ^5,6^ Ingredients of Concentrate 2 and 4: corn flakes, corn grain, wheat straw, barley flakes, white lupin flakes, barley seed, white lupin, pea flakes, hempseed meal, corn gluten meal, calcium carbonate, sodium chloride and vitamins. ^3,5^ Vitamin and mineral content (per kg): 37.21 mg of vitamin E; 0.47 mg of Co; 0.58 mg of I; 23.25 mg of Mn; 0.28 mg of Se. ^4,6^ Vitamin and mineral content (per kg): 39.90 mg of vitamin E; 0.50 mg of Co; 0.60 mg of I; 24.93 mg of Mn; 0.30 mg of Se.

**Table 3 animals-12-02922-t003:** Fatty acid profile (g/100 g of FAME ^1^) of the concentrates.

	CTR		HC	
Fatty Acid	Concentrate 1	Concentrate 3	Concentrate 2	Concentrate 4
C14:0	0.38	0.22	0.30	0.40
C16:0	21.08	15.50	14.44	16.22
C18:0	3.34	3.34	2.26	2.39
C20:0	0.86	0.62	0.67	0.74
C22:0	0.50	0.77	0.37	0.41
C24:0	0.45	0.41	0.33	0.38
Total SFA ^2^	27.69	21.29	18.78	21.11
C18:1n9	30.78	30.85	26.29	27.60
C20:1n9	0.48	0.42	0.47	1.00
C22:1n9	0.19	0.13	0.10	0.12
Total MUFA ^3^	32.35	32.28	27.68	28.99
C18:2n6	38.05	43.90	50.39	46.87
C18:3n6	0.04	0.03	0.19	0.09
C18:3n3	1.68	2.36	2.81	2.74
Total PUFA ^4^	39.97	46.42	53.54	49.89
Total n-6	38.19	44.01	50.69	47.09
Total n-3	1.77	2.41	2.85	2.81
n-6/n-3	21.5	18.2	17.8	16.8

^1^ FAME, fatty acid methyl esters; ^2^ SFA, saturated fatty acids; ^3^ MUFA, monounsaturated fatty acids; ^4^ PUFA.

**Table 4 animals-12-02922-t004:** Effect of dietary hemp cake inclusion on in-vivo performance of veal calves.

	Diet ^1^				*p*-Value
Item	CTR	HC	SEM ^3^	Diet	Pen
Initial BW, kg	59.7	59.7	0.75	0.883	<0.001
Final BW, kg	326.2	323.1	5.64	0.722	<0.001
ADG ^2^, kg/d	1.507	1.489	0.07	0.710	<0.001
Feed consumption					
Milk replacer, kg DM/d	1.241	1.241	--	--	--
Concentrate, kg DM/d	2.530	2.601	0.20	0.370	0.01 ^4^
Feed conversion ratio	2.502	2.533	0.04	0.332	--

^1^ Dietary treatments: CTR group: fed concentrate without hemp inclusion; HC group: fed concentrate with 3% hemp cake. ^2^ ADG = average daily gain. ^3^ SEM = standard error of least square means. ^4^ Instead of pen, feeders were used as block for concentrate consumption.

**Table 5 animals-12-02922-t005:** Effect of dietary hemp cake supplementation on carcass traits and physicochemical properties of *Longissimus dorsi* muscle at 7 days of storage of Holstein veal calves.

	Diet			
Item	CTR	HC	SEM	*p*-Value
Carcass parameters				
Carcass weight, kg	162.78	164.55	2.91	0.7644
Dressing percentage, % BW	49.91 ^b^	51.47 ^a^	0.24	0.0007
Thigh length, cm	67.92	67.81	0.64	0.8996
Rump width, cm	37.20 ^b^	38.38 ^a^	0.44	0.0496
pH	5.76	5.69	0.02	0.0663
Meat characteristics				
Lightness (L*)	45.83	47.34	0.46	0.1010
Redness (a*)	7.84	7.77	0.23	0.8834
Yellowness (b*)	14.95	14.90	0.16	0.8775
Hue angle (H*)	62.47	62.62	0.59	0.9060
Chroma (C*)	16.90	16.84	0.22	0.8964
Weight cooking losses (%)	29.49 ^b^	31.13 ^a^	0.38	0.0298
Shear force (N)	25.79 ^B^	36.18 ^A^	1.50	<0.0001
Water content (g/100 g muscle)	74.05	74.31	0.22	0.5847
IMF content (g/100 g muscle)	3.75	3.67	0.27	0.8911
Protein content (g/100 g muscle)	22.78	22.59	0.14	0.5060
Iron (mg/kg muscle)	2.83	2.95	0.09	0.5128

Dietary treatments: CTR group: fed concentrate without hemp supplementation; HC group: fed concentrate with hemp supplementation. IMF = Intramuscular Fat. ^a,b^ Mean values with different letters in superscript within rows indicate significant differences (*p* < 0.05); ^A,B^ Mean values with different letters in superscript within rows indicate significant differences (*p* < 0.01); SEM = standard error of least square means; *: the official abbreviation of parameters.

**Table 6 animals-12-02922-t006:** Effect of dietary hemp cake supplementation on fatty acid profile (g/100 g of total FA) of total lipids in the *Longissimus dorsi* muscle of Holstein veal calves.

	Diet			
Fatty Acid	CTR	HC	SEM	*p*-Value
C12:0	0.46	0.45	0.03	0.8632
C14:0	5.05	4.91	0.12	0.5377
C16:0	26.33	26.58	0.29	0.6835
C18:0	12.41	12.11	0.25	0.5490
Total SFA	45.48	45.07	0.39	0.6094
C14:1	0.67	0.72	0.03	0.3907
C16:1 n-7	3.35	3.25	0.07	0.4394
C18:1 n-9	38.94 ^b^	40.27 ^a^	0.34	0.0492
C18:1 t11	3.06 ^a^	2.71 ^b^	0.09	0.0466
Total MUFA	46.81	47.55	0.31	0.2403
C18:2 n-6	7.19	7.01	0.19	0.6608
C18:2 CLA	0.08	0.01	0.02	0.1243
C18:3 n-3	0.45 ^a^	0.36 ^b^	0.02	0.0299
Total PUFA	7.71	7.38	0.21	0.4345
n6:n3 ratio	16.77 ^b^	19.85 ^a^	0.66	0.0157

The concentration of fatty acids was expressed as g/100 g, considering 100 g the sum of the areas of all identified FAMEs. Dietary treatments: CTR group: fed concentrate without hemp supplementation; HC group: fed concentrate with hemp cake supplementation. IMF = Intramuscular Fat. ^a,b^ Mean values with different letters in superscript within rows indicate significant differences (*p* < 0.05); SEM = standard error of least square means. SFA = Saturated fatty acids; MUFA = Monounsaturated fatty acids; PUFA = Polyunsaturated fatty acids.

## Data Availability

Not applicable.

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
