# Peer review of "Effect of Dietary Hemp Cake Inclusion on the In Vivo and Post Mortem Performances of Holstein Veal Calves"

_animals, 2022, doi:10.3390/ani12212922_

Round 1

Reviewer 1 Report

The authors evaluated the effect of including hemp cake in the diet of Holstein calves. In particular, they evaluated the effect on growth performance and certain meat quality traits.

The topics covered are well within the aims of the journal. The manuscript is well written, the experimental scheme seems appropriate and the results and discussion are correctly reported. However, the authors reported that hemp supplementation did not influence the fatty acid profile of meat, but even if limited, some significant differences were observed.

Some suggestions are given in the attached file.

Reviewer 2 Report

The study is interesting and well written. The statistical analysis is appropriate. However, some results should be revised since SEM values related with ADG probably do not correspond with the p-value (Table 4). In table 5. p-value for dressing percentage should be re-checked. Remove trivial information from the discussion, lines 470-471 and line 474-475 are not conclusions. 

Reviewer 3 Report

Appreciate the authors for their study. 

In vivo – should be in italics

Why does the study consider only male animals?

Avoid multiple citations for a single sentence eg. Line no. 36 and it should be mentioned like [5-8]

Line no. 38 – Cited references [3, 8] not [3] [8]. – Revise it in the entire manuscript.

Abbreviations must be defined at first mention and used consistently thereafter in the entire manuscript eg. Line no. 41 PUFA, 47 – MUFA

Line no. 52: Cite only the numerical of reference.

There is no significant results of growth performance on final body weight, and FCR in HC group. Then how do authors claim that the hemp cake diet is effective?. As well, carcass weight was similar with the control and HC group, not to alter meat quality parameters

Line no. 332 – The sentence was correct? Related to the result?

Line no. 343: Farinon et al. [4], revise the same in the entire manuscript. Eg. Line no. 346, 371, 391

Round 2

Reviewer 3 Report

The authors revised well, in the revision.